# Reasoning with Memory Augmented Neural Networks for Language Comprehension

**Tsendsuren Munkhdalai & Hong Yu**
University of Massachusetts Medical School
Bedford VAMC
{tsendsuren.munkhdalai,hong.yu}@umassmed.edu

## Abstract

Hypothesis testing is an important cognitive process that supports human reasoning. In this paper, we introduce a new computational hypothesis testing framework that is based on memory augmented neural networks. Our approach involves a hypothesis testing loop that reconsiders and progressively refines a previously formed hypothesis in order to generate new hypotheses to test. We applied the proposed approach to language comprehension task by using Neural Semantic Encoders (NSE). Our NSE models achieved the state-of-the-art results showing an absolute improvement of 1.2% to 2.6% accuracy over previous results obtained by single and ensemble systems on standard machine comprehension benchmarks such as the Children's Book Test (CBT) and Who-Did-What (WDW) news article datasets.

## 1 Introduction

Formulating new hypotheses and testing them is a cognitive process that supports human reasoning and intelligence. This hypothesis testing process involves selective attention, working memory and cognitive control (Just and Carpenter, 1992; Polk and Seifert, 2002). Attention and working memory are engaged in order to maintain, manipulate, and update new hypotheses. Cognitive control is required to inspect and ignore incorrect hypotheses. Inspired by the hypothesis testing process in the human brain and to support dynamic reasoning of machines, in this work, we introduce a reasoning approach that is based on memory augmented neural networks (MANN). A new hypothesis is formed by regressing the original statement (i.e. query in context of QA). Then the hypothesis is tested against reality (i.e. data or document story). If the model is satisfied with the current test response or the hypothesis is true, the reasoning process is halted and the answer is found. Otherwise, another hypothesis is formulated by refining the previous one and the process is repeated until the answer is found.

While the idea of modeling hypothesis testing with MANN remains a generic reasoning framework and is applicable to several AI tasks, we apply this approach to cloze-type QA by using Neural Semantic Encoders (NSE). NSE is a flexible MANN architecture and have shown a notable success on several language understanding tasks ranging from sentence classification to language inference and machine translation (Munkhdalai and Yu, 2016). NSE has *read*, *compose* and *write* modules to manipulate external memories and it has introduced a concept of shared and multiple memory accesses, which has shown to be effective for sequence transduction problems.

Cloze-type question answering (QA) is a clever way to assess the ability of human and machine to comprehend natural language. This type of tasks are attractive for the natural language processing (NLP) community because the test sets or datasets can be generated without requiring expert's supervision, in an automatic manner, which is typically useful in training and testing artificial intelligent (AI) systems that can understand human language. In cloze-type QA setup, the machine is first presented with a text document containing a fact set and then it is asked to output answer for a query related to the document.

With the recent development of large-scale Cloze-type QA datasets (Hermann et al., 2015; Hill et al., 2015; Onishi et al., 2016) and deep neural network methods, remarkable advances have been made in order to solve the problem in an end-to-end fashion. The existing neural network approaches can

be broadly divided into single-step or multi-step comprehension depending on how documenting reading and answer inference processes are modeled. By mimicking human readers for a deeper reasoning, the multi-step comprehension systems have shown promising results (Hermann et al., 2015; Hill et al., 2015; Trischler et al., 2016; Sordoni et al., 2016). However, the current multi-turn models are designed with the predefined number of computational hops for inference while the difficulty of document and query pairs can vary. Some of the query-document pairs require a shallow reasoning like word or sentence level matching, but for some of them a deeper document level reasoning or a complex semantic understanding is crucial.

The proposed NSE comprehension models perform a reasoning process we called hypothesis-test loop. In each step, a new hypothesis for the correct answer is formed by a query regression. Then the hypothesis is checked and if it is true, the model halts the reasoning process (or the hypothesis-test loop) to give the correct answer. To this end, unlike previous methods with fixed computation our models introduce halting procedure in the hypothesis-test loop. When trained with classic back-propagation algorithm, our NSE models show consistent improvements over state-of-the-art baselines on two cloze-type QA datasets.

## 2  RELATED WORK

Recently, several large-scale datasets for machine comprehension have been introduced, including the cloze-type QA (Hermann et al., 2015; Hill et al., 2015; Onishi et al., 2016). Consequently there is an increasing interest in developing neural network approaches to solve the problem in an end-to-end fashion. The existing models for cloze-type QA can be categorized into single-step or multi-step approach depending on their comprehension process.

### 2.1  SINGLE-STEP COMPREHENSION

Singe-step comprehension methods read input document once with a single computational hop to make answer prediction. The reading process mainly involves context modeling with bi-directional recurrent neural networks (RNN) and selective focusing with attention mechanism. Hermann et al. (2015) introduced a CNN/Daily News QA task along with a set of baseline models such as Attentive Reader and Impatient Reader. The Attentive Reader model reads the document and the query with bi-directional LSTM (BiLSTM) networks and selects a query-relevant context by attending over the document. Chen et al. (2016) re-designed the Attentive Reader model (so called Stanford Attentive Reader) and examined the CNN/Daily News QA task. They found out that roughly 25% of all queries in Hermann et al. (2015)'s dataset is unanswerable and the recent neural network approaches have obtained the ceiling performance on this task. Kadlec et al. (2016) proposed an Attention Sum (AS) reader model that first attends over the document and then aggregates all the attention score for the same candidate answer to select the highest scoring candidate as correct answer.

However, for complex document and query pairs with deeper semantic association machine reading only once may not be enough and multiple reading and checking is crucial to perform deeper reasoning.

### 2.2  MULTI-STEP COMPREHENSION

Similar to a human reader, multi-step comprehension methods read the document and the query before making the final prediction. This kind of comprehension is mostly achieved by implementing an external memory and an attention mechanism to retrieve the query-relevant information from the memory throughout time steps. Hermann et al. (2015)'s Impatient Reader revisits the document states with the attention mechanism whenever it reads the next query word in each time scale. Hill et al. (2015) extended multi-hop Memory Networks (Sukhbaatar et al., 2015) with self-supervision to retrieve supporting memory. EpiReader performs a two-stage computation (Trischler et al., 2016). First it chooses a set of most probable answers with the AS model and forms new queries by replacing an answer placeholder in the original query with the candidate answer words. Second EpiReader runs an entailment estimation between the document and the new query pairs to predict the correct answer. Gated Attention (GA) reader extends the AS model with document-gating and iterative reading of the document and the query (Dhingra et al., 2016). The query representation is used as gating for the document and the iterative reading is accomplished by having separate bi-directional

gated recurrent unit (GRU) networks for each computational hop. Iterative Alternative Attention (IAA) reader (Sordoni et al., 2016) is a multi-step comprehension model which uses a GRU network to search for correct answers from a document. In this model, the GRU network is expected to collect evidence from the document and the query that assists prediction in the last time step. Cui et al. (2016) introduced attention-over-attention loss by computing a word level query-document matching matrix. This model provides a fine-grained word-level supervision signal which seems to help model training.

Our proposed model performs multiple computational steps for deeper reasoning. Unlike the previous work, in our model the number of steps to revise the document is not predefined and it is dynamically adapted for a particular document and query pair. Furthermore, we define novel ways to substitute query words with a word chosen from the document (i.e. regression process) and to check a hypothesis that the selected document word actually compliments the query (i.e. check process). When the hypothesis is true, our model halts the reading process and outputs the word chosen from the document as the correct answer. NSE is used as a controller for the whole process throughout the reasoning steps.

Among the aforementioned models, EpiReader seems to be the most relevant one to our language comprehension models. However, having an entailment estimation introduces a constraint in EpiReader which limits its application. Our model is generic and can be useful in different tasks other than machine comprehension such as language-based conversational tasks, knowledge inference and link prediction. As EpiReader has tightly integrated two-stage neural network modules, the performance directly depends on the first stage. If the first module misses out or fails to choose enough candidates, no correct answer can be found. Our model has no such issue and is not constrained in forming new queries. More recently, the idea of dynamic termination in the context of language comprehension is proposed by Shen et al. (2016) independently. While they use reinforcement learning, we explore two different approaches: adaptive computation and query gating to be fully trained by end-to-end back-propagation.

## 3   PROPOSED APPROACH

Our dataset consists of tuples $(D, Q, A, a)$, where $D$ is the document (i.e. passage) serving as a fact for the query $Q$, $A$ is a set of candidate answers and $a \in A$ is the true answer. The document and the query are sequences of tokens drawn from a vocabulary $V$. We train a model to predict the correct answer $a$ from the candidate set $A$ for given a pair of query and document.

The main components of our proposed model are shown in Figure 1. First the query and the document memory are initialized via context embedding (omitted from the figure). The memories are then processed with memory read and write operations throughout the hypothesis-test loop. In each step of the loop, the read module formulates a new hypothesis by updating the query memory with relevant content from the document memory. Then the write module tests whether the new hypothesis is true by inspecting the current query and the document states. Selecting relevant content from the document to regress the old query is essentially an inference (i.e. prediction) in our model. Intuitively the input query is regressed toward (becoming) a complete query containing the correct answer word within it. If the write module thinks that the query is complete and the correct answer is found, it halts the hypothesis-test loop. The write module also supervises the query state transitions and retains the right to roll back the new query changes during the reasoning process. To avoid an overconfident prediction and to halt the reasoning process, we explore two different strategies: (a) query gating and (b) adaptive computation.

Like a human reader, the model reads the document multiple times, formulates a hypothetical answer for the query and tests it against the story of the document throughout the hypothesis-test steps. Once satisfied with the response of the current hypothesis, the model outputs the response as the correct answer.

### 3.1   MEMORY INITIALIZATION

Instead of using raw word embeddings, we initialize the document and the query memories via context embedding in order to inform the memory slots of contextual information from text passages.

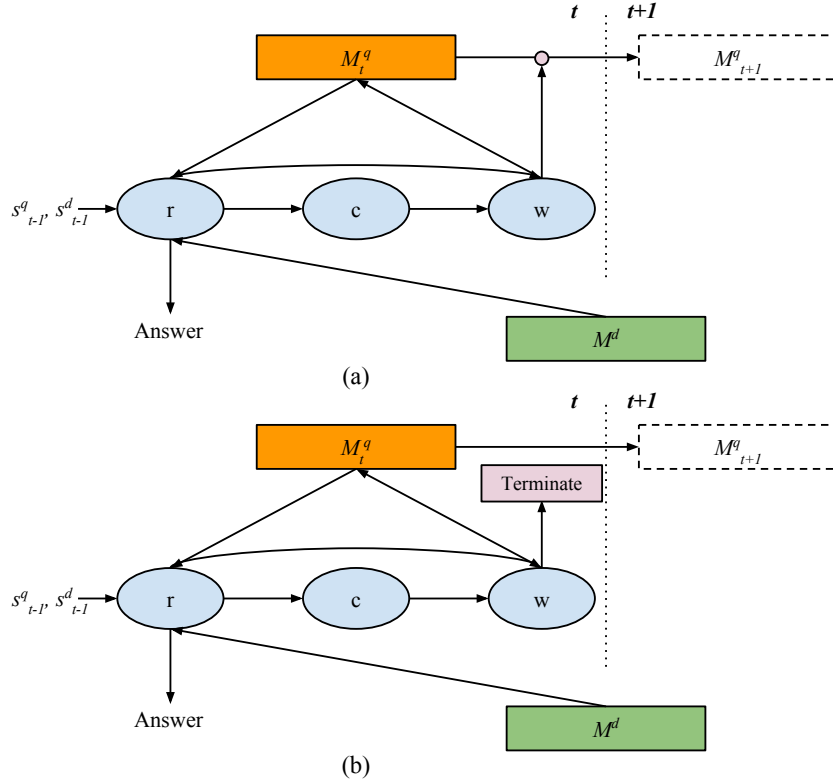

Figure 1: High-level architectures of our proposed models: the NSE Query Gating model (a) and the NSE Adaptive Computation model (b). The query memory of the former is gated to the next step by the write module whereas the query memory in the latter is updated and passed to the next step without gating and the write module is trained to halt the hypothesis-test loop. r: read, c: compose and w: write module.

Two BiLSTM networks are applied to the document and the query sequences separately, as:

$$M_0^q = BiLSTM^q(Q) \tag{1}$$

$$M^d = BiLSTM^d(D) \tag{2}$$

where $M_0^q \in R^{k \times |Q|}$ and $M^d \in R^{k \times |D|}$ are the memory representations. $k$ is the size of BiLSTM hidden layer. The query memory is evolved over time while the document memory is not updated and rather serves as a fact set for the query.

Note that in our model the context embedding network can be any network accepting word embeddings as input, such as multilayer perceptron (MLP) or convolutional neural network. We choose BiLSTM because it is able to learn the word-centered context representation effectively by reading text with LSTMs from both left and right and concatenating resulting hidden vectors.

## 3.2 HYPOTHESIS TESTING

The query and the document memories are further processed through an iterative process called hypothesis-test loop. In each step of the loop, the query memory $M^q$ is updated with content from the document memory to form a new query (i.e. *hypothesis formulation*). The new query is then checked against the document facts and used to make an answer prediction (i.e. *hypothesis testing*). The NSE read, compose and write modules collectively perform the following overall process.

**Read:** this module takes in the query and the document states $s_{t-1}^q, s_{t-1}^d \in R^k$ from the previous time step $t-1$ as input and computes the alignment vectors $l_t^q \in R^{|Q|}, l_t^d \in R^{|D|}$, the new query and the document states $s_t^q, s_t^d \in R^k$ and the query memory key vector $z_t^q \in R^{|Q|}$ as follows [1]:

$$r_t = read^{LSTM}([s_{t-1}^q; s_{t-1}^d]) \tag{3}$$

$$l_t^q = r_t^\top M_{t-1}^q \tag{4}$$

$$s_t^q = softmax(l_t^q)^\top M_{t-1}^q \tag{5}$$

$$z_t^q = sigmoid(l_t^q) \tag{6}$$

$$l_t^d = s_t^{q\top} M^d \tag{7}$$

$$s_t^d = softmax(l_t^d)^\top M^d \tag{8}$$

Intuitively depending on the previously retrieved document content as well as the previous query states, the read module retrieves from the document a word to be relocated to the query and computes the positions of this word in the query. Since the document word can be located in multiple different positions in the query, $sigmoid$ function is used to normalize the alignment vector $l_t^q$. The document state $s_t^d$ represents the newly retrieved word and the key vector $z_t^q$ defines its new position in the query memory. Because such a decision is made sequentially in every step, we equip the read module with a LSTM network (i.e. $read^{LSTM}$). We initialize the document and the query states $s_0^d, s_0^q$ with the last hidden states of the query and the document BiLSTM networks.

**Compose:** the compose module combines the current query and document states $s_t^q, s_t^d$ and the current hidden state of the read module $r_t \in R^k$ as:

$$c_t = compose^{MLP}(s_t^q, s_t^d, r_t) \tag{9}$$

The resulting single vector $c_t \in R^k$ is passed to the write module for subsequent process. The compose module can be viewed as a feature extractor from the current document and query pair. By taking in the hidden state $r_t$, the compose module also informs the write module of the read module's current decision.

**Write:** this module accepts the outputs of the read module and updates the query memory as follows:

$$M_t^q = M_{t-1}^q z_t^q + s_t^d(\mathbf{1} - z_t^q) \tag{10}$$

where $\mathbf{1}$ is a matrix of ones. Note that the values of the key vector $z_t^q$ ranges from zero to one and zero (or near zero) values indicate the query position where the new document word is written.

The write module is also responsible for checking the new hypothesis in order to decide whether to halt the hypothesis-test loop for the final answer or to continue. We explore two different strategies to be discussed below. Both methods take in the output of the compose module and employs the LSTM to make the sequential decision.

### 3.2.1 QUERY GATING

Figure 1 (a) shows the overall architecture of our model with query gating mechanism. In this model instead of making a hard decision on halting the loop (i.e. stop reading), the write module performs a word-level query gating as:

$$w_t = write^{LSTM}(c_t) \tag{11}$$

$$g_t^q = sigmoid(w_t^\top M_{t-1}^q) \tag{12}$$

$$M_t^q = M_t^q(\mathbf{1} - g_t^q) + M_{t-1}^q g_t^q \tag{13}$$

where $\mathbf{1}$ is a matrix of ones. The key part in the above equation is obtaining the gating weights. This is accomplished by comparing each memory slot with the hidden vector $w_t \in R^k$ and normalizing

---

[1]For brevity, we omit the broadcasting operations from the equations.

Table 1: Statistics of the datasets. train (s): train strict, train (r): train relaxed and cands: candidates.

| | WDW | | | | CBT-NE | | | CBT-CN | | |
|---|---|---|---|---|---|---|---|---|---|---|
| | train (s) | train (r) | dev | test | train | dev | test | train | dev | test |
| # queries | 127,786 | 185,978 | 10,000 | 10,000 | 108,719 | 2,000 | 2,500 | 120,769 | 2,000 | 2,500 |
| avg. # cands | 3.5 | 3.5 | 3.4 | 3.4 | 10 | 10 | 10 | 10 | 10 | 10 |
| avg. # tokens | 365 | 378 | 325 | 326 | 433 | 412 | 424 | 470 | 448 | 461 |
| vocab size | 308,602 | | 347,406 | | 53,063 | | | 53,185 | | |

resulting scores with $sigmoid$ function. Therefore, the gating vector $g_t^q \in R^{|Q|}$ have ones for preserving and zeros for erasing content of the old query.

This can be seen as a memory gating process which prevents the model from forgetting the old query information. Note that even if the query memory is updated with the document content given by the read module, the write module makes the final decision based on features extracted by the compose module. In other words if the write module decides to keep the old query information, the changes in the new query are simply ignored and the same query from the previous time step is passed along to the next step. The number of steps $T$ in the hypothesis-test loop is a hyperparameter in this model. Therefore, in this setup the write module is expected to lock the query state with its gating mechanism as soon as the hypothesis is true.

### 3.2.2 ADAPTIVE COMPUTATION

In this model, the write module is equipped with a termination head as shown in Figure 1 (b). Particularly, the write module with the termination head decides its willingness to continue or finish the computation in each step.

We define a probabilistic framework for halting. Our approach is similar to the input and output handling mechanism of Neural Random-Access Machines (Kurach et al., 2016). In each time step, the write module outputs a termination score $e_t$ as follows:

$$w_t = write^{LSTM}(c_t) \tag{14}$$

$$e_t = sigmoid(o^\top w_t) \tag{15}$$

where $o \in R^k$ is a trainable vector that projects the hidden state $w_t$ to a scalar value. Then the probability to halt the hypothesis-test loop after $t$ steps is

$$p_t = e_t \prod_{i=1}^{t-1}(1 - e_i) \tag{16}$$

We also introduce a hyperparameter $T$ for the maximum number of permitted steps. If the model runs out of the time without halting the process (after $T$ steps), we force the model to output the final answer in step $T$. In this case, the probability to stop reading is

$$p_T = 1 - \sum_{i=1}^{T-1} p_i \tag{17}$$

### 3.3 ANSWER PREDICTION

In step $t$, the query-to-document alignment score $l_t^d$ is used to compute the probability $P(a|Q, D)$ that the answer $a$ is correct given the document and the query. In particular, we adapt the pointer sum attention mechanism (Kadlec et al., 2016) as

$$P_t(a|Q, D) = v^\top softmax(l_t^d) \tag{18}$$

where $v \in R^{|D|}$ is a mask denoting the positions of the answer token $a$ in the document (ones for the token and zeros otherwise).

Table 2: Model comparison on the CBT dataset.

| Model | CBT-NE | | CBT-CN | |
|---|---|---|---|---|
| | dev | test | dev | test |
| Human (context + query) (Hill et al., 2015) | - | 81.6 | - | 81.6 |
| LSTMs (context + query) (Hill et al., 2015) | 51.2 | 41.8 | 62.6 | 56.0 |
| MemNNs (window mem. + self-sup.) (Hill et al., 2015) | 70.4 | 66.6 | 64.2 | 63.0 |
| AS Reader (Kadlec et al., 2016) | 73.8 | 68.6 | 68.8 | 63.4 |
| GA Reader (Dhingra et al., 2016) | 74.9 | 69.0 | 69.0 | 63.9 |
| EpiReader (Trischler et al., 2016) | 75.3 | 69.7 | 71.5 | 67.4 |
| IAA Reader (Sordoni et al., 2016) | 75.2 | 68.6 | 72.1 | 69.2 |
| AoA Reader (Cui et al., 2016) | 77.8 | 72.0 | 72.2 | 69.4 |
| MemNN (window mem. + self-sup. + ensemble) (Hill et al., 2015) | 70.4 | 66.6 | 64.2 | 63.0 |
| AS Reader (ensemble) (Kadlec et al., 2016) | 74.5 | 70.6 | 71.1 | 68.9 |
| EpiReader (ensemble) (Trischler et al., 2016) | 76.6 | 71.8 | 73.6 | 70.6 |
| IAA Reader (ensemble) (Sordoni et al., 2016) | 76.9 | 72.0 | 74.1 | 71.0 |
| NSE ($T = 1$) | 76.2 | 71.1 | 72.8 | 69.7 |
| NSE Query Gating ($T = 2$) | 76.6 | 71.5 | 72.3 | 70.7 |
| NSE Query Gating ($T = 6$) | 77.0 | 71.4 | 73.0 | **72.0** |
| NSE Query Gating ($T = 9$) | 78.0 | 72.6 | 73.5 | 71.2 |
| NSE Query Gating ($T = 12$) | 77.7 | 72.2 | **74.3** | 71.9 |
| NSE Adaptive Computation ($T = 2$) | 77.1 | 72.1 | 72.8 | 71.2 |
| NSE Adaptive Computation ($T = 12$) | **78.2** | **73.2** | 74.2 | 71.4 |

For the query gating model, we use the probability $P_T(a|Q, D)$ produced in the last step $T$ to choose the correct answer. For the second model, we incorporate the termination score $p_t$ and redefine the probability as

$$P(a|Q, D) = \sum_{i=1}^{T} (p_i \cdot P_i(a|Q, D)) \tag{19}$$

We then train the models to minimize cross-entropy loss between the predicted probabilities and correct answers.

## 4 EXPERIMENTS

We evaluated our models on two large-scale datasets: Childrens Book Test (CBT) (Hill et al., 2015) and Who-Did-What (WDW) (Onishi et al., 2016). We focused on these tasks because there is still a large gap between the human and the machine performances on CBT and WDW, which is in contrast to the CNN/Daily News QA datasets covered in Section 2. The CBT dataset was constructed from the children book domain whereas the WDW corpus was built from the news article domain (the English Gigaword corpus); therefore we think that the two datasets are quite representative for evaluation of our models. Furthermore the CBT dataset comes with two difficult tasks depending on the type of answer words to be predicted: named entity (CBT-NE) and common nouns (CBT-CN). The WDW training set has two different setups with strict and relaxed baseline suppression. Table 1 summarizes some important statistics of the datasets.

### 4.1 TRAINING DETAILS

We chose one-layer LSTM networks for the read and the write modules and an MLP with single-layer for the composition module. We used stochastic gradient descent with an Adam optimizer to train the models. The initial learning rate ($lr$) was set to 0.0005 for CBT-CN or 0.001 for other tasks. A pre-trained 300-D Glove 840B vectors (Pennington et al., 2014) were used to initialize the word embedding layer[2]; therefore the embedding layer size is 300. The hidden layer size of the context embedding BiLSTM nets $k = 436$. The embeddings for out-of-vocabulary words and the model parameters were randomly initialized from the uniform distribution over [-0.1, 0.1). The gradient

---

[2]http://nlp.stanford.edu/projects/glove/

Table 3: Model comparison on the WDW dataset.

| Model | Strict | | Relaxed | |
|---|---|---|---|---|
| | dev | test | dev | test |
| Human (Onishi et al., 2016) | - | 84.0 | - | - |
| Attentive Reader (Hermann et al., 2015) | - | 53.0 | - | 55.0 |
| AS Reader (Kadlec et al., 2016) | - | 57.0 | - | 59.0 |
| GA Reader (Dhingra et al., 2016) | - | 57.0 | - | 60.0 |
| Stanford Attentive Reader (Chen et al., 2016) | - | 64.0 | - | 65.0 |
| NSE ($T = 1$) | 65.1 | 65.5 | 66.4 | 65.3 |
| NSE Query Gating ($T = 2$) | 65.4 | 65.1 | 65.7 | 65.5 |
| NSE Query Gating ($T = 6$) | 65.5 | 65.7 | 65.6 | 65.8 |
| NSE Query Gating ($T = 9$) | 65.8 | 65.8 | 65.8 | 65.9 |
| NSE Query Gating ($T = 12$) | 65.2 | 65.5 | 65.7 | 65.4 |
| NSE Adaptive Computation ($T = 2$) | 65.3 | 65.4 | 66.2 | 66.0 |
| NSE Adaptive Computation ($T = 12$) | **66.5** | **66.2** | **67.0** | **66.7** |

clipping threshold was set to 15. The models were regularized by applying 20% dropouts to the embedding layer[3]. We used the batch size $n = 32$ for the CBT dataset and $n = 25$ for the WDW dataset and early stopping with a patience of 1 epoch. For the WDW dataset, we anonymized the answer candidates by following the work of Onishi et al. (2016) and Hermann et al. (2015).

We run a hyperparameter search over $k = \{256, 368, 436, 512\}$, $lr = \{0.0005, 0.001\}$ and $l_2$ $decay = \{0.0001, 0.00005, 0.00001\}$ on the CBT dev sets to come up with the current setting for training. Among these parameters, the $l_2$ weight decay regularizer did not seem to help and thus it was not applied. We did not tune the dropouts.

We used the following batching heuristic in order to speedup the training. We created a temporary example pool by randomly sampling from the training set and sorted them according to the length of the document. Then the first $n$ examples ($n = 32$ or $n = 25$) of the ordered pool were put into the same batch, without replacement, and the rest of the pool was replaced back in the training set. This is performed until there is not enough training examples to create a new pool. Finally the documents in the same batch are padded with special symbol $<pad>$ to the same length. The batches were regenerated in every epoch to prevent the model from learning a simple mapping function. We also padded queries in the same batch with the special symbol to an equal length.

## 4.2 RESULTS

In Tables 2 and 3 we compared the performance of our models with all published models (as baselines) including their ensemble variations. We report the performance of our query gating model at the varying number of hypothesis-test steps $T = \{1, 2, 6, 9, 12\}$. $T$ for our adaptive computation model was set to 2 or 12. When $T = 1$, both models update the memory only once and do not have enough time to read it back and thus they are reduced to the same model called NSE (i.e. NSE $T = 1$). Typically when $T$ is greater than one, the proposed halting strategies result in two different models.

Our query gating model achieves 72.6% and 72.0% accuracy on the CBT tasks outperforming all the previous baselines. The performance varies for the different number of steps. For the larger number of allowed steps, the query gating model tends to overfit on the CBT dataset as the performance gap between the dev and test sets increases. Our NSE Adaptive Computation model sets the best score on CBT-NE task with 73.2% accuracy. Overall our NSE models bring modest improvements of 1.2% on CBT-NE and 2.6% on CBT-CN tasks and now the performance differences between the human and the machines are only 8.4% and 9.6%.

On the WDW task our single model, NSE with adaptive computation, scores 2.2% and 1.7% higher than the previous best result of Stanford Attentive Reader. The NSE Query Gating model obtains its best result when $T = 9$. Comparing to the proposed variations of NSE, our Adaptive Computation

---

[3]More detail on hyperparameters can be found in our code: https://bitbucket.org/tsendeemts/nse-rc

model is more robust and sets new state-of-the-art performance on three out of four tasks by effectively deciding when to halt the reasoning processing with the termination head. Memory locking (to prevent from forgetting) did not show to be as effective as the termination-based approach on this task. In Appendix A, we visualize query regression process in both models and observe that while new queries are generated, query words relevant to the correct answer are not overwritten by the memory write module.

The performance of the NSE model with $T = 1$ matches that of the previous single systems. In general given a small number of permitted steps, the proposed models tend to less overfit but the final performance is not high. As the number of permitted steps increases both dev and test accuracy improves yielding an overall higher performance. However, this holds up to a certain point and with a large permitted steps we no longer observe a significant performance improvement in terms of testing accuracy. For example, NSE Query Gating model with $T = 15$ (not included in the table) achieved 79.2% dev accuracy and 71.4% test accuracy on CBT-NE task, showing the highest accuracy on the dev set yet massively overfitting on the test set. Furthermore it becomes expensive to train a model with a large number of allowed steps.

It is worth noting that even if the proposed models in this work were trained using classic end-to-end back-propagation, they can easily be trained with reinforcement learning methods, such as the REINFORCE algorithm (Williams, 1992), for further evaluation. Such an adaptation is particularly straightforward for NSE Adaptive Computation because this model has already incorporated a probabilistic termination head in it.

## 5 CONCLUSION

Inspired by cognitive process of hypothesis testing in human brain, we proposed a reasoning approach based memory augmented neural networks and applied it to language comprehension. Our proposed NSE models with dynamic reasoning have achieved the state-of-the-art results on two machine comprehension tasks. In order to halt reasoning process we explored two different strategies to be fully trained with classic back-propagation algorithm: query memory gating which prevents from forgetting old query and adaptive computation with termination head. The NSE adaptive computation model has shown to be effective in the experiments. Our proposed models can be trained using reinforcement learning. We plan to apply our approach to other AI tasks, such language-based conversational tasks, link prediction and knowledge inference.

### ACKNOWLEDGMENTS

We would like to thank Abhyuday Jagannatha, Jesse Lingeman and the reviewers for their insightful comments and suggestions. This work was supported in part by the grant HL125089 from the National Institutes of Health and by the grant 1I01HX001457-01 supported by the Health Services Research & Development of the US Department of Veterans Affairs Investigator Initiated Research. Any opinions, findings and conclusions or recommendations expressed in this material are those of the authors and do not necessarily reflect those of the sponsor.

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

# A    QUERY REGRESSION ANALYSIS

We analyzed how input queries are updated through memory read, write and gating operations in order for the model to make a prediction for the correct answer.

## A.1    QUERY GATING

Figure 2 depicts the memory key ($z_t^q$) and the query gate ($g_t^q$) states of the NSE Query Gating model for the input document shown on the top. As noted previously, the model updates the query memory at a position that has a lower memory key value (i.e. the values in the memory key closer to zero indicate the update positions) but rolls back or ignores the new query change if the gating value is one at the same position. The top attended words resulted by the memory read operation define the information for the update. We listed the top-3 attended words in Figure 2, which already includes the correct answer.

Note how the the memory key and the gating states are changed during the regression steps. The memory key values are varied across the query positions. Overall the query tokens 'He' and 'Meadows' are pointed to be updated. As proceeding with the query regression, the model effectively adjusts its gates. During the first two steps, the model is willing to accept new updates occurred at certain positions such as 'He' and 'Meadows'. However, the gating value reaches one and the gates are closed in the later steps. The model no longer accepts new query updates in step 6 and is now ready to output the correct answer.

Our analysis showed that the model rarely updates the query at the place holder position (i.e. 'XXXXX' in the query). This is desirable in such iterative query regression process. Here the placeholder information remains unchanged in order to be used in the subsequent steps and to inform the model of what part of the query needs to be completed.

## A.2    ADAPTIVE COMPUTATION

In Figure 3, we showed the memory key and the termination head states for an input pair from the CBT dev set. The model actively performs the query regression during the first two steps and then it halts the process as the termination head reaches one in the third step. Interestingly, the memory key points to the most of the query positions for update in the first step and then its values gradually increase in the later steps implying no information exchange between the document and the query. The answer cue word 'Johnny' and the place holder information are not overwritten until the loop is halted.

**Correct answer:** Green
**Document:** retorted Chatterer. ``I know enough to know when I am well off.`` Who has a discontented heart Is sure to play a sorry part." Johnny **Chuck** crawled out of the old log and stretched himself somewhat painfully.`` That may be, but there are different kinds of discontent. Who never looks for better things Will live his life in little rings. Well, I must be moving along, if I am to see the world. "So Johnny **Chuck** bade Chatterer good-by and started on. It was very delightful to wander over the **Green** Meadows on such a beautiful spring morning. The violets and the wind-flowers nodded to him, and the dandelions smiled up at him. Johnny almost forgot his torn clothes and the bites and scratches of his great fight with the gray old **Chuck** the day before. It was fun to just go where he pleased and not have a care in the world. He was thinking of this, as he sat up to look over the **Green** Meadows. His heart gave a great throb. What was that over near the lone **elm**-tree? It was -- yes, it certainly was another **Chuck**! Could it be the old gray **Chuck** come back for another fight? A great anger filled the heart of Johnny **Chuck**, and he whistled sharply. The strange **Chuck** didn't answer. Johnny ground his teeth and started for the lone **elm**-tree.
**Top-3 attended words:** 1)Green 2)Chuck 3)elm

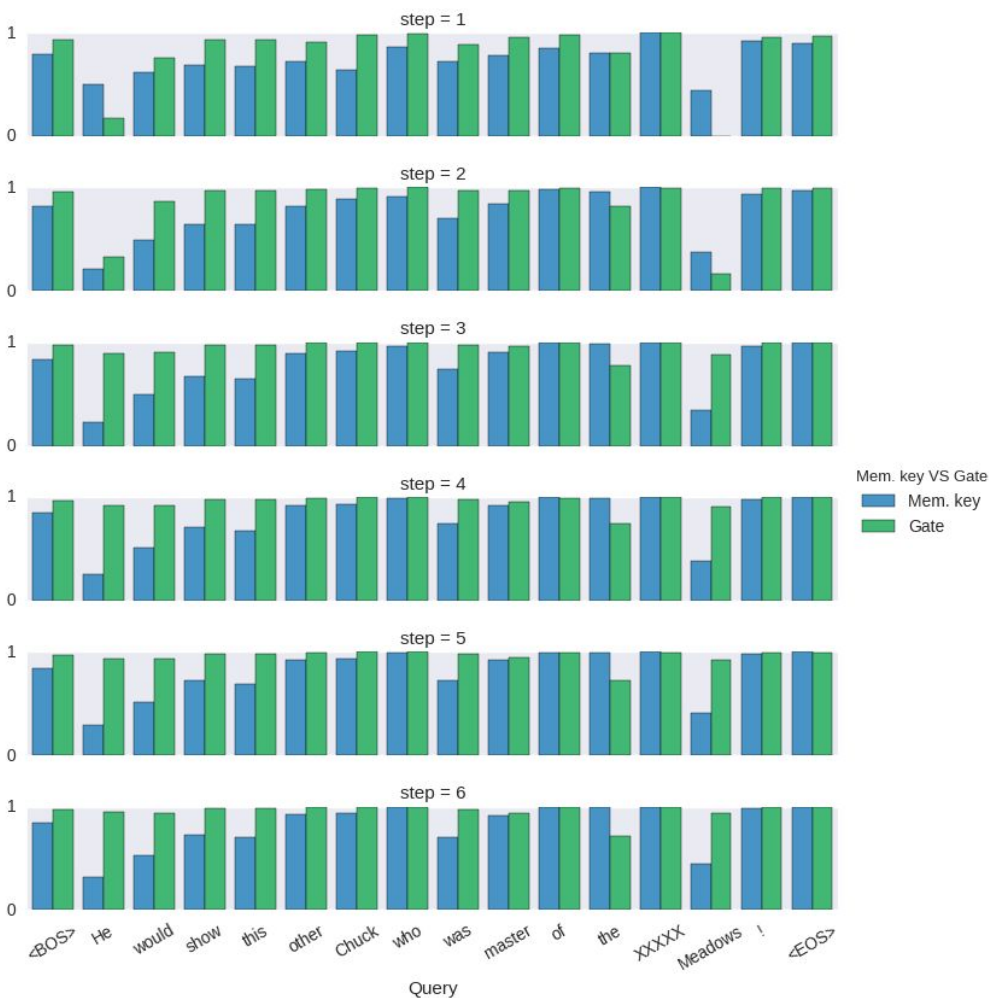

Figure 2: Visualization of query regression process in the NSE query gating model. On the top, input document along with the correct answer is shown. The top-3 attended words throughout the hypothesis-testing loop are highlighted. On the bottom, we visualize the states of the memory key $z_t^q$ and the query gate $g_t^q$. The hypothesis-testing steps $t$ ranging from 1 to 6 are only shown due to the space limitation. The gating value near one is to ignore a new query update while the memory key value closer to zero indicates a new query update position.

**Correct answer:** Chuck

**Document:** No, it looked like the shadow of Johnny **Chuck**. **Peter** rubbed his eyes and looked again. Then he hurried as fast as he could, lipperty-lipperty-lip. The nearer he got, the less like Johnny **Chuck** looked the one sitting on Johnny **Chuck**'s door-step. Johnny **Chuck** had gone to sleep round and fat and roly-poly, so fat he could hardly waddle. This fellow was thin, even thinner than **Peter Rabbit** himself. He waved a thin hand to **Peter**. ``Hello, **Peter Rabbit**! I told you that I would see you in the spring. How did you stand the long winter? "That certainly was Johnny **Chuck**'s voice. **Peter** was so delighted that in his hurry he fell over his own feet.`` Is it really and truly you, Johnny **Chuck**?' 'he cried. ``Of course it's me; who did you think it was? "replied Johnny **Chuck** rather crossly, for **Peter** was staring at him as if he had never seen him before. ``I -- I -- I didn't know," confessed **Peter Rabbit**. ``I thought it was you and I thought it wasn't you. What have you been doing to yourself, Johnny **Chuck**? Your coat looks three sizes too big for you, and when I last saw you it didn't look big enough."

**Top-3 attended words:** 1)Chuck 2)Rabbit 3)Peter

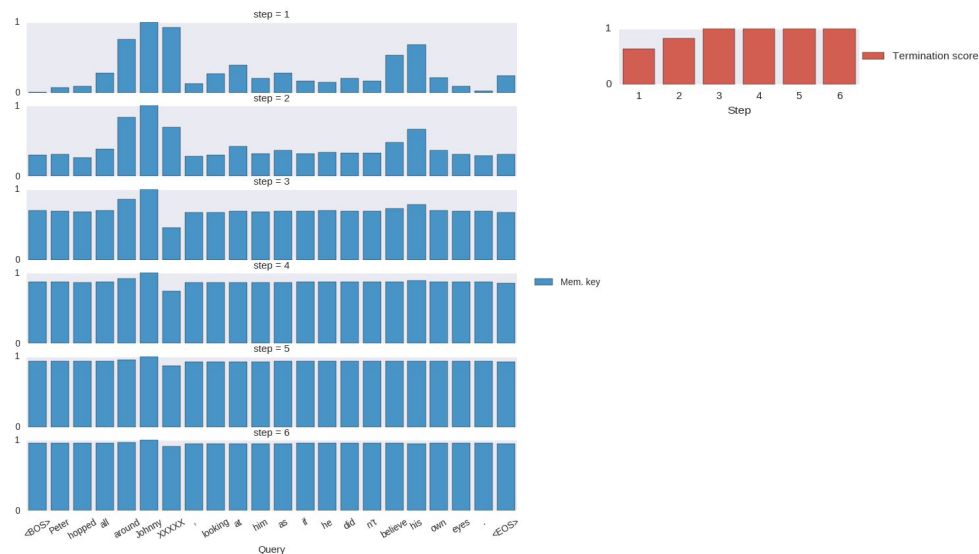

Figure 3: Visualization of query regression process in the NSE adaptive computation model. On the top, input document along with the correct answer is shown. The top-3 attended words throughout the hypothesis-testing loop are highlighted. On the bottom, we visualize the states of the memory key $z_t^q$ and the termination head. The memory key value closer to zero indicates a new query update position and the termination score closer to one is to halt the hypothesis testing loop.

