# Peer review of "Reasoning with Memory Augmented Neural Networks for Language Comprehension"

_ICLR 2017 — accepted_

[Official Review · AnonReviewer3 · rating 7 · confidence 4 · 17 Dec 2016]
**A novel and interesting approach**

This paper proposed an iterative query updating mechanism for cloze-style QA. The approach is novel and interesting and while it is only verified in the paper for two Cloze-style tasks (CBT and WDW), the concept of read/compose/write operations seem to be more general and can be potentially applied to other reasoning tasks beyond Cloze-style QA. Another advantage of the proposed model is to learn when to terminate the iteration by the so-called adaptive computation model, such that it avoids the issue of treating the number of iterations as another hyper-parameter, which is a common practice of iterative models/multi-hop reasoning in previous papers.

There are a couple places that this paper can improve. First, I would like to see the results from CNN/Daily Mail as well to have a more comprehensive comparison. Secondly, it will be useful to visualize the entire M^q sequence over time t (not just z or the query gating) to help understand better the query regression and if it is human interpretable.

[Official Review · AnonReviewer1 · rating 6 · confidence 3 · 21 Dec 2016]
**No Title**

Thie paper proposed an iterative memory updating model for cloze-style question-answering task. The approach is interesting, and result is good. For the paper, I have some comments:

1. Actually the model in the paper is not single model, it proposed two models. One consists of "reading", "writing", "adaptive computation" and " Answer module 2", the other one is "reading", "composing", "writing", "gate querying" and "Answer module 1". Based on the method section and the experiment, it seems the "adaptive computation" model is simpler and performs better. And without two time memory update in single iteration and composing module, the model is similar to neural turing machine.

2. What is the MLP setting in the composing module? 

3. This paper tested different size of hidden state:[256, 368, 436, 512], I do not find any relation between those numbers, how could you find 436? Is there any tricks helping you find those numbers?

4. It needs more ablation study about using different T such as T=1,2..

5. According to my understanding, for the adaptive computation,  it would stop when the P_T <0. So what is the distribution of T in the testing data?

[Official Review · AnonReviewer2 · rating 7 · confidence 2 · 28 Dec 2016]
**Shows improvement on state of the art; complexity ?**

First I would like to apologize for the delay in reviewing.

Summary : this work introduces a novel memory based artificial neural network for reading comprehension. Experiments show improvement on state of the art.
The originality of the approach seems to be on the implementation of an iterative procedure with a loop testing that the current answer is the correct one.

In order to get a better sense of the reason for improvement it would be interesting to have a complexity and/or a time analysis of the algorithm. I might be mistaken but I don't see you reporting anything on the actual number of loops necessary in the reported experiments.

The dataset description in section 2.2, should be moved to section 4 where the other datasets are described.

[Author Response · Tsendsuren Munkhdalai · 13 Jan 2017]
**Conducted additional experiments**

1) We performed query regression analysis and added visualizations in the appendix.
2) We conducted additional experiments of 12 different runs when T=1,2 and included the results and discussion.

[Public Comment · yelong shen · 21 Jan 2017]
**Connection with ReasoNet Model ?**

The proposed model in the paper is very similar to the ReasoNet model[1], which is first public available in arXiv in Sep, 17th, 2016. 

1. The organization of the paper, especially motivation and related work work part, is very similar to ReasoNet paper [1].
2. The idea of termination Gate in Comprehension task is first proposed in ReasoNet, the paper fails to explain the connection between this work and ReasoNet model [1].

1. Yelong Shen, Po-Sen Huang, Jianfeng Gao, Weizhu Chen. "ReasoNet: Learning to Stop Reading in Machine Comprehension. arXiv preprint arXiv:1609.05284 (Sep-17-2016).

[Final Decision · Program Chairs · 06 Feb 2017]
**ICLR committee final decision**

This paper proposes a memory-enhanced RNN in the vein of NTM, and a novel training method for this architecture of cloze-style QA. The results seem convincing, and the training method is decently novel according to reviewers, although the evaluation seemed somewhat incomplete according to reviewers and my own reading. For instance, it is questionable whether or not the advertised human performance on CNN/DM is accurate (based on 100 samples from a 300k+ dataset), so I'm not sure this warrants not evaluating or reporting performance on it. Overall this looks like an acceptable paper, although there is room for improvement.